# Impact of Health Literacy on Frailty among Community-Dwelling Seniors

**DOI:** 10.3390/jcm7120481

**Published:** 2018-11-26

**Authors:** Chi Hsien Huang, Yu-Cheng Lai, Yi Chen Lee, Xiao Tong Teong, Masafumi Kuzuya, Kuang-Ming Kuo

**Affiliations:** 1Department of Community Healthcare and Geriatrics, Nagoya University Graduate School of Medicine, Nagoya 4668550 Japan; evaairgigaa@gmail.com (C.H.H.); kuzuya@med.nagoya-u.ac.jp (M.K.); 2Department of Family Medicine, E-Da Hospital, I-Shou University, Kaohsiung City 82445, Taiwan; ed107116@edah.org.tw; 3School of Medicine for International Students, I-Shou University, Kaohsiung City 82445, Taiwan; 4Department of Nutrition Therapy, E-DA Hospital, Kaohsiung City 82445, Taiwan; ed103549@edah.org.tw; 5Adelaide Medical School, The University of Adelaide, Adelaide, South Australia 5005, Australia; xiaotong.teong@adelaide.edu.au; 6South Australian Health and Medical Research Institute (SAHMRI), Adelaide, South Australia 5001, Australia; 7Institutes of Innovation for Future Society, Nagoya University, Nagoya 4648601, Japan; 8Department of Healthcare Administration, I-Shou University, Kaohsiung City 82245, Taiwan

**Keywords:** health literacy, frailty, elderly

## Abstract

Health literacy has been reported to have effects on health behavior change and health-related outcomes, but few studies have explored the association between health literacy and frailty. The aim of our study is to investigate the relationships between health literacy and frailty among community-dwelling seniors. This cross-sectional study enrolled 603 community-dwelling older adults (307 women) in residential areas, with a mean age of 70.9 ± 5.82 years. Health literacy was assessed using the Mandarin version of the European Health Literacy Survey Questionnaire. Physical frailty was defined by Fried frailty phenotype. Logistic regression was carried out to determine potential risk factors of frailty. In the multivariate logistic regression model, physical activity (Odds Ratio [OR] 1.47, 95% Confidence Interval [CI] 1.06–2.03) and health literacy (sufficient vs. excellent: OR 2.51, 95% CI 1.32–4.77) were associated with prefrailty and frailty. In subgroup analysis, pre-frailty and frailty were also negatively associated with health literacy in individuals with ‘insufficiently active’ (inadequate vs. excellent: OR 5.44, 95% CI 1.6–18.45) and ‘sufficiently/highly active’ physical activity levels (sufficient vs. excellent: OR 2.41, 95% CI 1.07–5.42). Therefore, in these community-dwelling elderly adults, health literacy was associated with pre-frailty and frailty regardless of age, gender, socio-economic status, and education level.

## 1. Introduction

Frailty is a clinical geriatric syndrome in the elderly characterized by high vulnerability and low resilience [1] This geriatric giant is associated with an increased likelihood of worse health outcomes including falls, disability, hospitalization, in institutionalization and mortality [2,3,4,5]. Moreover, frailty has been demonstrated to be related to stress, life satisfaction and quality of life [6]. Owing to the complex domains of physical, psychological, socio-economic, and environmental factors linked to frailty, a comprehensive multi-disciplinary and multi-step intervention approach including nutritional support and physical exercise is recommended for treating frail older populations [1]. However, due to poor awareness of frailty itself and prevention strategies for the elderly, the initial step in frailty management is promoting knowledge and improving self-awareness [7,8].

Recently, health literacy, defined as the ability to read, understand, evaluate, and use health information to make reasoned, health-related decisions [9], has been reported to be associated with health behavior and outcomes among adults with chronic diseases such as obesity, cardiovascular disease, and diabetes [10,11]. No studies have explored the association between health literacy and frailty, except for one cross-sectional study which did not control for several confounding factors including diseases, prescription drugs, protein intake and physical activity [12]. Considering that the current strength and quality of evidence is insufficient, more studies are needed to clarify these relationships. Thus, we designed the present study to determine whether health literacy is associated with frailty in community-dwelling older adults.

## 2. Experimental Section

### 2.1. Study Design

This cross-sectional study was conducted from February to October 2017 in Kaohsiung City in southern Taiwan. We recruited participants by sending invitation letters and posting handbills at nine community centers located in three residential areas characterized as >3000, 300–3000, and <300 people per square kilometer for urban, suburban, and rural areas, respectively. Only individuals older than 65 years old who could provide informed consent were eligible for inclusion and further investigation by a trained nurse. Those who had impairments in the Barthel index of activities of daily living, were unable to perform a 5-m walk test, had active cancers or incurable diseases with an estimated life expectancy of 6 months or less, and those who could not complete an interview owing to severe hearing or visual impairment were excluded. The study protocol was approved by the Institutional Review Board of E-Da Hospital (EMRP-105-79).

### 2.2. Measures 

Baseline characteristics were acquired through interviews using questionnaires. Age, gender, body mass index (BMI), education level, annual income (none, ≤8000 USD, >8000 USD), medical history and medication lists were obtained. Daily protein and calorie intake were assessed using a face-to-face interview with food frequency questionnaire conducted by a trained nurse. The information was confirmed by 24-h dietary recall conducted by a dietician via a telephone interview. Physical activity was evaluated using the Taiwanese version of the International Physical Activity Questionnaire (IPAQ), and was further categorized into highly active, sufficiently active, and insufficiently active groups [13]. Geriatric depression scale-5 (GDS-5), a five-item questionnaire, was used to identify depression by a cut-off of 2 points [14].

Health literacy was assessed using the validated Mandarin version of the European Health Literacy Survey Questionnaire [15,16]. Three dimensions of health literacy including health care health literacy (HCHL), disease prevention health literacy (DPHL), and health promotion health literacy (HPHL) composed a 47-item self-reported questionnaire (HLS-EU-Q47) [17]. To facilitate clinical practice and enhance community awareness, a cross-validated 16-item short form (HLS-EU-Q16) has been developed [18]. We used the Mandarin version of the HLS-EU-Q16 and answers were coded as follows: 1 = very difficult, 2 = difficult, 3 = easy, and 4 = very easy. To reduce ambiguous responses, the option ‘don’t know’ was excluded from our version. The total sum score was generated and converted to a 0–50-point index scale. The level of health literacy was thus categorized into four groups of inadequate (score 0 to 25), problematic (score 26 to 33), sufficient (score 34 to 42), and excellent (score 43 to 50 points) health literacy groups [17].

In this study, frailty was defined as fulfilling three out of five phenotypic criteria: unintentional weight loss, fatigue, slow walking speed, low physical activity, and weakness consistent with those used in the Cardiovascular Health Study (CHS) [5]. Individuals meeting one or two criteria were defined as having a pre-frail status, and those who met none of the criteria were defined as being robust. A loss of ≥3 kilograms or ≥5% of body weight in the past year was defined as unintentional weight loss [19]. Exhaustion was identified by two statements from the Center for Epidemiological Studies-Depression (CES-D) scale: (a) I felt that everything I did was an effort; (b) I could not get going [20]. The positive criterion is defined as the presence of at least one condition for three days or more during the last week [5]. Slow gait speed was defined as the lowest quintile in a 5-m walking speed test without acceleration and deceleration stratified by gender and height. In men, the cutoff points of those whose heights were ≤168 and >168 cm were 0.89 and 0.96 m/s, respectively; while corresponding values in women whose heights were ≤156 and >156 cm were 0.85 and 0.88 m/s, respectively. Low physical activity was defined as the lowest quintile of activity level assessed using the Taiwanese version of the IPAQ [13]. The following cutoff values stratified by gender were used: ≤1236 kcal/week for men, ≤1212 kcal/week for women. Weakness was defined by low handgrip strength as measured by a digital dynamometer (TTM-YD, Tokyo, Japan). The cut-off values of handgrip strength across increasing quintiles stratified by gender and BMI (≤22.6, 22.61–24.8, 24.81–26.7, >26.7 kg/m^2^) were 20.9, 21.6, 22, and 22.5 kg in men, and 13, 15.4, 16.8, and 16.4 kg in women, respectively.

### 2.3. Statistical Analysis

We analyzed data using IBM SPSS Statistics for Windows, Version 25.0. (IBM Corp., Armonk, NY, UK). First, demographic data including number, percentage, mean and standard deviation were summarized by gender. Comparisons between men and women were based on an analysis of variance (for continuous variables) and on a chi-square test (for dichotomous variables). A *p*-value of 0.05 was used to determine statistical significance. A logistic regression model was used to determine the associations between health literacy and prefrailty and frailty. Participant characteristics and potential variables were entered in a simultaneous model with adjustments.

## 3. Results

### 3.1. Characteristics of the Participants 

The mean age of all respondents (*n* = 603) was 70.9 ± 5.82 years, with men being on average older than the women (Table 1). There were significant differences in the level of education and health literacy between male and female participants. Although more of the men (36.15%) had an education level above junior high school than the women (22.74%), slightly more women had sufficient and excellent health literacy (44.62%) compared to men (42.92%) (Figure 1). Total daily calorie and protein intake were 20.25 ± 4.97 kcal/kg/day and 0.66 ± 0.19 g/kg/day, respectively (Table 1). With respect to physical activity and anthropometric measures, men were more physically active, had a faster walking speed and greater handgrip strength than women (Table 1). With regards to frailty, 4.05% and 19.32% of the men had pre-frailty and frailty respectively, compared to 1.95% and 54.72% of women. Additional descriptive statistics are shown in Table 1.

On the other hand, participants with better health literacy were younger and highly educated and had higher income, fewer morbidities, less polypharmacy, and less depressive moods. Moreover, they were more physically active, with faster walking speed and greater hand grip strength. Detailed descriptive profiles are shown in Table 2.

### 3.2. Relationships among Health Literacy, Pre-Frailty, and Frailty

The univariate logistic regression model showed that education, health literacy, multiple comorbidities, GDS-5 scores, and physical activity were significantly associated with pre-frailty and frailty (Table 3). In the multivariate logistic regression model, physical activity (OR 1.47, 95% CI 1.06–2.03) and health literacy (sufficient vs. excellent: OR 2.51, 95% CI 1.32–4.77) remained significantly associated with prefrailty and frailty after adjusting for age, sex, BMI, education level, annual income, multiple comorbidities, polypharmacy, depression, and protein intake (Table 3). Individuals with ‘insufficiently active’ physical activity levels were at a higher risk of having pre-frailty and frailty (OR 1.55, 95% CI 1.07–2.23) compared to those with ‘sufficiently/highly active’ physical activity (Table 3).

In subgroup analysis by physical activity, pre-frailty and frailty were negatively associated with annual income (0 vs. >8000 USD: OR 5.44, 95% CI 1.01–29.17) and health literacy (sufficient vs. excellent: OR 4.12, 95% CI 1.28–13.22; inadequate vs. excellent: OR 5.44, 95% CI 1.6–18.45) in individuals with ‘insufficiently active’ physical activity (Table 4). On the other hand, for individuals with ‘sufficiently/highly active’ physical activity, sufficient health literacy was negatively correlated with pre-frailty and frailty (OR 2.41, 95% CI 1.07–5.42) (Table 4). A high BMI was associated with a higher risk of having pre-frailty and frailty (OR 1.11, 95% CI 1.02–1.21) in those with sufficiently/highly active physically active (Table 4).

## 4. Discussion

This is one the first studies to investigate the association between health literacy and frailty in community-dwelling healthy elderly. Our results showed that low health literacy was associated with a non-robust status including pre-frailty and frailty. Therefore, aside from nutrition and exercise habits, health literacy seems to be another modifiable factor for frailty interventions.

The relationship between health literacy and frailty may potentially be explained by behavior change theory. Although frailty interventions including nutritional support and exercise programs have been shown to be effective [21], participation and adherence to intervention programs play key roles in improving outcomes. Owing to the current low adherence rate to interventions, it is necessary to emphasize the importance of maintaining behavioral changes [22]. Health literacy has been shown to result in positive reinforcement in seeking information, changing attitudes, and behavioral changes [9], and it has been broadly investigated and associated with the long-term outcomes of smoking cessation, diabetes control, and medication adherence [23,24,25]. However, the relationship between health literacy and frailty has seldomly been reported. Our findings show that health literacy was an independent predictor of pre-frailty and frailty after controlling for education level, which is consistent with another cross-sectional study [12]. By increasing motivation and encouraging autonomy, health literacy has been shown to improve long-term adherence to frailty management strategies [26].

Of note, our results suggest that sufficient health literacy may be not be sufficient to prevent frailty in the elderly. In the subgroup analysis of physical activity, lower health literacy in the participants with insufficient physical activity was associated with an increased risk of pre-frailty and frailty. Although the trend was not significant in the sufficiently/highly physically active group, sufficient health literacy still remained a protective factor for pre-frailty and frailty. Therefore, the implementation of health literacy promotion appears to be an important issue.

From the viewpoint of stakeholders and policy makers, it has been proposed that frailty management should involve increasing awareness and understanding of frailty among the general public [27,28]. In addition, an integrated individualized knowledge translation strategy has been suggested as the initial step to implement frailty prevention and management in the community [28]. Considering the perception gap in community-dwelling elderly, promoting health literacy may both improve knowledge and also reinforce a positive attitude and adherence to interventional approaches. In a weight reduction intervention program, improving health literacy was demonstrated to have a positive influence on dietary intake behavior, but not on physical activity habits [29]. Therefore, future intervention studies including health literacy promotion are warranted to investigate the effects on elderly subjects with different levels of health literacy.

Several health literacy measures such as the Rapid Estimate of Adult Literacy in Medicine (REALM) [30], the Test of Functional Health Literacy in Adults (TOFHLA) [31], and the Newest Vital Sign (NVS) [32] have been validated for busy primary care settings. However, these performance-based measures, which are highly time-consuming and examiner-dependent, may be not appropriate for the community settings. On the other hand, HLS-EU-Q which is subjective health-decision based has been concurrently validated with TOFHLA in a population-based sample [33]. To promote frailty prevention for the aging population in the community, HLS-EU-Q seems to be more applicable and appropriate.

There are several limitations to this study. This was a cross-sectional study, and thus causal relationships could not be determined. In addition, owing to the convenience sampling method, the proportion of frailty was lower in our study (2.99%) than in another cohort study conducted in elderly community-dwelling Taiwanese [34]. However, the incidence of pre-frailty was similar in both studies. The results implied that our target population was relatively less frail and had greater access to health services. In other words, excluding the physically disabled and individuals with impaired activities of daily life may have underestimated the frailty status in the community. Moreover, participants with advanced old age or terminal illness were not recruited in the present study. This may mean that our findings cannot be applied to the whole elderly population. Despite these limitations, due to the complexity and difficulty in eradicating frailty, promotion of frailty prevention is of importance and clinical relevance and worth intervening as early as possible. Health literacy can modify the process of understanding and self-awareness of disease and might be a facilitator to improve the effects of frailty intervention. Therefore, our study has potential to trigger more studies to explore the influence of health literacy on frailty management for the aging population.

## 5. Conclusions

This study shows that health literacy is associated with frailty regardless of age, gender, socio-economic status, and education level in a population of community-dwelling older adults from Taiwan. To enhance recognition of frailty and improve frailty management, health literacy should be considered as a modifiable factor for frailty intervention strategies

## Figures and Tables

**Figure 1 jcm-07-00481-f001:**
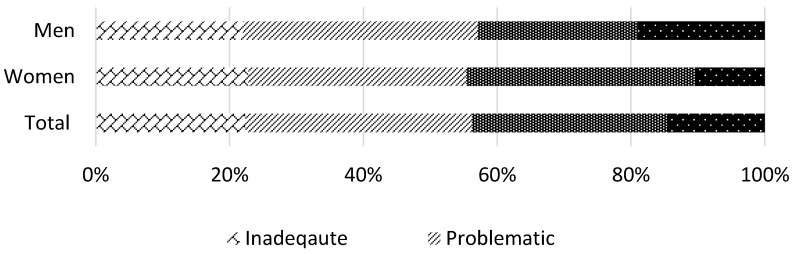
Level of health literacy by sex.

**Table 1 jcm-07-00481-t001:** Characteristics of the participants by sex

Characteristics	Total (*n* = 603)	Men (*n* = 296)	Women (*n* = 307)	*p* Value *
**Age, years (mean ± standard deviation [SD])**	70.9 ± 5.82	72.04 ± 6.45	69.81 ± 4.90	**<0.001**
BMI, kg/m^2^ (mean ± SD)	24.67 ± 2.99	24.52 ± 2.84	69.81 ± 4.10	0.23
**Education level, *n* (%)**				
Illiterate	128 (21.23%)	43 (14.53%)	85 (27.69%)	**<0.001**
Elementary school	299 (49.59%)	146 (49.32%)	153 (49.84%)	
Junior and senior high school	130 (21.56%)	70 (23.65%)	60 (19.54%)	
College or above	46 (7.63%)	37 (12.5%)	9 (2.93%)	
**Health literacy, *n* (%)**				
Inadequate	135 (22.4%)	65 (22.0%)	70 (22.8%)	**0.004**
Problematic	204(33.83%)	104 (35.14%)	100 (32.57%)	
Sufficient	176 (29.19%)	71 (24%)	105 (34.2%)	
Excellent	88 (14.59%)	56 (18.92%)	32 (10.42%)	
Annual income, *n* (%)				
No income	46 (7.63%)	17 (5.74%)	29 (9.45%)	0.08
≤8000 USD	389 (64.51%)	187 (63.18%)	202 (65.8%)	
>8000 USD	168 (27.86%)	92 (31.08%)	76 (24.75%)	
Multiple comorbidities, No. of diseases (%)				
0–2	521 (86.4%)	257 (86.82%)	264 (86%)	0.77
≥3	82 (13.6%))	39 (13.18%)	43 (14%)	
Polypharmacy, No. of medications (%)				
<5	183 (30.35%)	93 (31.42%)	90 (29.32%)	0.57
≥5	420 (69.65%)	203 (68.58%)	217 (70.68%)	
GDS-5 score				
<2	483 (80.1%)	238 (80.41%)	245 (79.8%)	0.85
≥2	120 (19.9%)	58 (19.59%)	62 (20.2%)	
**Physical activity (IPAQ), *n* (%)**				
Insufficiently active	281 (46.6%)	132 (44.59%)	149 (48.53%)	**0.02**
Sufficiently active	284 (47.1%)	137 (46.28%)	147 (47.88%)	
Highly active	38 (6.3%)	27 (9.12%)	11 (3.58%)	
Calorie intake, kcal/kg/day (mean ± SD)	20.25 ± 4.97	19.97 ± 4.36	20.51 ± 5.48	0.18
Macronutrients				
Carbohydrate intake, g/kg/day	3.01 ± 0.85	2.97 ± 0.78	3.06 ± 0.92	0.17
Fat intake, g/kg/day	0.61 ± 0.2	0.61 ± 0.18	0.62 ± 0.21	0.3
Protein intake, g/kg/day	0.66 ± 0.19	0.66 ± 0.17	0.67 ± 0.20	0.87
**Walking speed, m/s (mean ± SD)**	1 ± 0.2	1.03 ± 0.17	0.97 ± 0.22	**<0.001**
**Handgrip, kg (mean ± SD)**	23.46 ± 7.74	28.14 ± 7.45	18.95 ± 4.75	**<0.001**
Frailty status, *n* (%)				
Robust	271(44.94%)	138 (46.62%)	133 (43.32%)	0.18
Pre-frailty	314 (52.07%)	146 (49.32%)	168 (54.72%)	
Frailty	18 (2.99%)	12 (4.05%)	6 (1.95%)	

GDS-5, Geriatric depression scale-5; IPAQ, International Physical Activity Questionnaires. * Comparison of mean value of the characteristics between men and women.

**Table 2 jcm-07-00481-t002:** Characteristics of the participants by health literacy (HL).

Characteristics	Inadequate HL (*n* = 135)	Problematic HL (*n* = 204)	Sufficient HL (*n* = 176)	Excellent HL (*n* = 88)	*p* Value *
**Age, years (mean ± SD)**	73.2 ± 6.1	71.1 ± 6.1	70.0 ± 4.9	68.5 ± 5.0	**<0.001**
BMI, kg/m^2^ (mean ± SD)	24.6 ± 3.1	24.7 ± 3.2	24.8 ± 2.9	24.4 ± 2.4	0.71
**Education level, *n* (%)**					
Illiterate	56 (41.5%)	46 (22.5%)	23 (13.1%)	3 (3.4%)	**<0.001**
Elementary school	68 (50.4%)	126 (61.8%)	86 (48.9%)	19 (21.6%)	
Junior and senior high school	11 (8.1%)	30 (14.7%)	63 (35.8%)	26 (29.5%)	
College-or-above	0	2 (1.0%)	4 (2.3%)	40 (45.5%)	
**Annual income, *n* (%)**					
No income	3 (2.2%)	14 (6.9%)	28 (15.9%)	1 (1.1%)	**<0.001**
≤8000 USD	83 (61.5%)	152 (74.5%)	110 (62.5%)	44 (50.0%)	
>8000 USD	49 (36.3%)	38 (18.6%)	38 (21.6%)	43 (48.9%)	
**Multiple comorbidities, No. of diseases (%)**					
0–2	99 (73.3%)	178 (87.3%)	159 (90.3%)	85 (96.6%)	**<0.001**
≥3	36 (26.7%)	26 (12.7%)	17 (9.7%)	3 (3.4%)	
**Polypharmacy, No. of medication (%)**					
<5	21 (15.6%)	58 (28.4%)	60 (34.1%)	44 (50.0%)	**<0.001**
≥5	114 (84.4%)	146 (71.6%)	116 (65.9%)	44 (50.0%)	
**GDS-5 scores**					
<2	80 (59.3%)	165 (80.9%)	156 (88.6%)	82 (93.2%)	**<0.001**
≥2	55 (40.7%)	39 (19.1%)	20 (11.4%)	6 (6.8%)	
**Physical activity (IPAQ), *n* (%)**					
Insufficiently active	94 (69.9%)	88 (43.1%)	61 (34.1%)	38 (43.2%)	**<0.001**
Sufficiently active	38 (28.1%)	100 (49.0%)	101 (57.4%)	45 (51.1%)	
Highly active	3 (2.2%)	16 (7.8%)	14 (8.0%)	5 (5.7%)	
Calorie intake, kcal/kg/day (mean ± SD)	20.4 ± 4.6	20.4 ± 5.6	19.8 ± 4.9	20.4 ± 4.2	0.62
Macronutrients					
Carbohydrate intake, g/kg/day	3.0 ± 0.7	3.1 ± 1.0	3.0 ± 0.8	3.0 ± 0.8	0.56
Fat intake, g/kg/day	0.6 ± 0.2	0.6 ± 0.2	0.6 ± 0.2	0.6 ± 0.2	0.17
Protein intake, g/kg/day	0.7 ± 0.2	0.7 ± 0.2	0.6 ± 0.2	0.7 ± 0.2	0.23
**Walking speed, m/s (mean ± SD)**	1.0 ± 0.2	1.0 ± 0.2	1.0 ± 0.2	1.1 ± 0.2	**<0.001**
**Handgrip, kg (mean ± SD)**	23.4 ± 6.5	23.3 ± 7.3	21.6 ± 8.4	27.8 ± 7.5	**<0.001**
**Frailty status, *n* (%)**					
Robust	46 (34.1%)	105 (51.5%)	64 (36.4%)	56 (63.6%)	**<0.001**
Pre-frailty	84 (62.6%)	93 (45.6%)	106 (60.2%)	31 (35.2%)	
Frailty	5 (3.7%)	6 (2.9%)	6 (3.4%)	1 (1.1%)	

GDS-5, Geriatric depression scale-5; IPAQ, International Physical Activity Questionnaires. * Comparison of mean value of the characteristics between participants with different health literacy.

**Table 3 jcm-07-00481-t003:** Risk factors for pre-frail and frail status by odds ratio in logistic regression analysis

	Pre-Frailty and Frailty (Versus Robust)							
Variable	Univariate Model				Multivariate Model *			
	Odds Ratio	95% CI		*p* Value	Odds Ratio	95% CI		*p* Value
		Lower Limit	Upper Limit			Lower Limit	Upper Limit	
Age (years)	1.03	1.00	1.06	0.07	1.02	0.98	1.05	0.33
Sex								
Female	1.00				1.00			
Male	0.88	0.63	1.21	0.42	1.06	0.74	1.52	0.76
BMI (kg/m^2^)	1.05	0.99	1.11	0.09	1.04	0.98	1.11	0.18
**Education level**								
Illiterate	2.84	1.42	5.71	**<0.01**	1.66	0.65	4.24	0.29
Elementary school	2.34	1.23	4.45	**0.01**	1.46	0.63	3.41	0.38
Junior/senior high school	1.56	0.78	3.10	0.21	0.95	0.42	2.18	0.91
College or above	1.00				1.00			
**Health literacy**								
Inadequate	3.39	1.93	5.94	**<0.01**	1.71	0.82	3.56	0.15
Problematic	1.65	0.99	2.76	0.06	1.11	0.58	2.14	0.74
Sufficient	3.06	1.80	5.21	**<0.01**	2.51	1.32	4.77	**0.01**
Excellent	1.00				1.00			
Annual income								
No income	1.92	0.97	3.82	0.06	1.52	0.72	3.22	0.28
≤8000 USD	1.14	0.79	1.64	0.48	0.99	0.65	1.51	0.96
>8000 USD	1.00				1.00			
**Comorbidities** **(No. of diseases)**								
0–2	1.00				1.00			
≥3	1.91	1.16	3.14	**0.01**	1.35	0.78	2.32	0.28
Polypharmacy (No. of medications)								
<5	1.00				1.00			
≥5	1.40	0.99	1.99	0.06	1.02	0.69	1.52	0.92
**GDS-5 score**								
<2	1.00				1.00			
≥2	1.92	1.26	2.93	**<0.01**	1.58	0.99	2.52	0.053
**Physical activity (IPAQ)**								
Insufficiently active	1.47	1.06	2.03	**0.02**	1.55	1.07	2.23	**0.02**
Sufficiently/highly active	1.00				1.00			
Protein intake (g/kg/day)								
<0.8	1.15	0.52	2.55	0.72	1.06	0.45	2.53	0.89
0.8–1	0.71	0.30	1.68	0.44	0.70	0.28	1.73	0.44
≥1	1.00				1.00			

GDS-5, Geriatric depression scale-5; IPAQ, International Physical Activity Questionnaires. * Adjusted R squared = 0.115.

**Table 4 jcm-07-00481-t004:** Adjusted odds ratios for pre-frail and frail status by physical activity

	Pre-Frailty and Frailty (Versus Robust)							
Variable	Insufficiently Physically Active Category ^#^				Sufficiently/Highly Physically Active Category *			
	Odds Ratio	95% CI		*p* Value	Odds Ratio	95% CI		*p* Value
		Lower Limit	Upper Limit			Lower Limit	Upper Limit	
Age (years)	0.99	0.94	1.04	0.59	1.04	0.99	1.09	0.09
Sex								
Female	1.00				1.00			
Male	1.51	0.87	2.65	0.15	0.75	0.44	1.26	0.28
**Body weight index [BMI] (kg/m^2^)**	1.02	0.92	1.13	0.68	1.11	1.02	1.21	**0.02**
Education level								
Illiterate	1.05	0.22	4.99	0.95	1.67	0.49	5.73	0.42
Elementary school	1.90	0.45	7.95	0.38	0.79	0.26	2.39	0.67
Junior/senior high school	1.13	0.29	4.49	0.86	0.62	0.21	1.86	0.39
College or above	1.00				1.00			
**Health literacy**								
Inadequate	5.44	1.60	18.45	**0.01**	0.56	0.19	1.69	0.31
Problematic	2.62	0.81	8.51	0.11	0.73	0.32	1.68	0.46
Sufficient	4.12	1.28	13.22	**0.02**	2.41	1.07	5.42	**0.03**
Excellent	1.00				1.00			
**Annual income**								
No income	5.44	1.01	29.17	**0.048**	0.98	0.37	2.57	0.96
≤8000 USD	0.98	0.54	1.81	0.96	1.00	0.52	1.94	0.99
>8000 USD	1.00				1.00			
Multiple comorbidities (No. of diseases)								
0–2	1.00				1.00			
≥3	1.04	0.49	2.19	0.93	2.03	0.85	4.84	0.11
Polypharmacy (No. of medications)								
<5	1.00				1.00			
≥5	0.59	0.31	1.15	0.12	1.44	0.84	2.47	0.19
GDS-5 score								
<2	1.00				1.00			
≥2	1.63	0.82	3.25	0.17	1.71	0.86	3.41	0.13
Protein intake (g/kg/day)								
<0.8	2.22	0.68	7.27	0.19	0.35	0.08	1.44	0.15
0.8–1	1.01	0.28	3.65	0.99	0.27	0.06	1.14	0.07
≥1	1.00				1.00			

GDS-5, Geriatric depression scale-5. ^#^ Adjusted R squared = 0.176. * Adjusted R squared = 0.17.

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
