# Peer review of "Impact of Health Literacy on Frailty among Community-Dwelling Seniors"

_jcm, 2018, doi:10.3390/jcm7120481_

Round 1
Reviewer 1 Report
This study sets itself the task of establishing a relationship between two entities net of the influence of other related entities. It takes some expense in measurement and data analysis to check the available evidence so that a positive answer can be given. The two entities in this case are health literacy and frailty. I have described this endeavor in this somewhat peculiar way to make the point that the goal of this study is a fairly simple one. One can always demand that more and more evidence be presented before one dares to make conclusion, but I think there is an imbalance between the presentation of analyses in voluminous tables and their contribution to the core finding.
The methodology, that is the measures and analyses, have been chosen with care and oversight, and the analysis is presented expertly. There is one exception, though. The choice of the HLS-EUQ16 as a measure of health literacy I consider unfortunate. This measure, the shorter as well as the longer version employs situational items and asks respondents to indicate how easy or how difficult it is for them to make a decision or act in a particular way in these situations. These replies in my view are much more an indicator of the claim to have a say in decisions relevant to one’s health than of an objective assessment or performance-based measure of a person’s capability of doing that. Those performance-based measures exist in the domain of health literacy, the most widely used is, just to name one, the S-TOFHLA. The HLS-EUQ16 measure, as it has been used in this study, can therefore be said to target health empowerment more than health literacy, which is of course a serious objection against the bottom line of this analysis.
The manuscript almost always shies away from attempts at making sense of the findings, and if it does not, the assertions quickly lose argumentative power. That becomes visible as frailty is more or less treated as a regular disease that is subject to prevention if one is ready to try that. This view is summarized in the last sentence of the discussion: “… it may be of importance to measure and follow-up health literacy in the elderly to prevent and delay the occurrence of frailty.”Not many would agree that observing health literacy has the potential to prevent frailty.
In this context I miss a discussion of how death and advanced old age and incapacity to take part in the study may have affected the results.
For the editors I think the decision is one between solid craftsmanship in conducting such research and a doubtful relevance of the analysis.
Minor points:
Abstract, line 25, “fried” must be a typo
Methods, line 82, the HLS-EU scale is not a Likert scale (Likert’s is a scale that measures attitudes and attitude strength at the same time, usually by different levels of (dis)agreement)
Discussion, line 35, typo causal/casual
Reviewer 2 Report
Overall an interesting study of an aspect of aging, frailty and its relation to health literacy. Some specific issues to be resolved/revised:
Line 38 - "modern geriatric giant" - if this is referring to frailty a rather subjective characterization and potentially overstated and untrue - frailty has always been part of aging and health risks Why do the authors think this is a modern problem?
Line 43: reasonably the sentence could end with defining precisely whom these recommendations are aimed at - are these recommendations only those considered "the elderly" or is the advice appropriate for those classified as "aging populations" (especially if the goal is prevention)?
Line 93: what was the content of the two statements by the Ctr for Epidemiological Studies and how did these statements allow investigators to assess exhaustion? more information is needed
Lines 115-116: "and the men were older than the women" might be revised as "with men being on average (or generally) older than the women"
Page 11 Line 2: "Pioneer" study sounds awkward in English. Perhaps the authors should revise to note that this is one of the first (or the first) study to investigate the associations between HL and frailty
Lines 14-15: "Our findings showed that health literacy was an independent predictor of pre-frailty and frailty regardless of education level" seems to contradict a previous comment on page 3, line 125 that "participants with better health literacy were younger and highly educated". This seeming inconsistency needs to be addressed.
LInes 40-41: The authors note a limitation of the study which is commendable, however without measuring those with physical disabilities the findings will NOT, as the authors suggest, be generalizable to the public nor will it actually move the science forward in a rigorous manner. The results are suggestive of a relationship but hardly conclusive without a true measure of the public that does include those with disabilities.
Round 2
Reviewer 1 Report
The authors have very carefully and comprehensively reacted to my comments to the earlier version of the paper. The reactions are mainly additions to the discussion section, as it deals with presenting their views on the use of the HLS-EU, the relevance of their research in light of the efficacy in attempting to prevent frailty and possible bias that may enter the sampling as the very old, the very frail and the very sick have to be excluded. I did not expect these points could be changed, but saw a need for better explanations. These the authors have provided.